# OpenReview forum: "Do Vision-Language Models Respect Contextual Integrity in Location Disclosure?"
_ICLR.cc/2026/Conference — ICLR 2026 Poster_

### Official Review · Reviewer_RmhQ · 2025-10-21

**Soundness:** 2
**Presentation:** 3
**Contribution:** 3
**Rating:** 4
**Confidence:** 4

**Summary:**

This paper introduces VLM-GeoPrivacy, a benchmark designed to test whether VLMs respect contextual integrity when disclosing image locations.
It evaluates 14 leading VLMs on 1,200 real-world images annotated for context, intent, and appropriate disclosure granularity, showing that current models frequently over-disclose sensitive locations and fail to align with human privacy norms.
Even advanced models like GPT-5 and Gemini-2.5 achieve only about 50% agreement with human judgments and are easily manipulated by iterative or adversarial prompts.
The authors conclude that future multimodal systems must incorporate context-aware privacy reasoning rather than blanket disclosure limits to protect users from location-based privacy risks.

**Strengths:**

+ Evaluate many frontier models, providing a comprehensive view.
+ The framing of the paper is good. The motivation is clear. I like the idea that “models should be aware of whether the geo-information in specific images can be revealed.” The framing about three possible use cases in section 2.1 is good.
+ The questions are constructed based on some existing regulations. The authors used an iterative design of annotation for better refinement.

**Weaknesses:**

- The conversion from the Nissenbaum’s Contextual Integrity and some established regulations to Q1-7 lacks some supports. [See my Q1]
- Some designs look ad-hoc: Phi-3.5 for filtering, GPT-4o-mini for labeling, GPT-4.1-mini for the judge. [See my Q2]
- As the dataset is the core part of this paper. Information about annotation process should be clearer. [See my Q3]

**Questions:**

1. Line 155-156: “we found that adding these concrete, intermediate questions improves annotator consistency.” I want to know that how much do you think that the previous questions are deliberate guidance to annotators towards a desired result? The Krippendorff’s alpha = 0.83 is good. Do you test the alpha when Q1-6 are not given? I think people’s intuition should also be considered. Another possible test can be first letting annotators do Q7, then inviting them to give explanations using Q1-6 to see the contributions of Q1-6 in human decisions.
2. Do you have human verification on Phi-3.5-Vision filtering results? A better way may be using multiple VLMs and do majority voting? Do you use GPT-4o-mini for classifying images to the several privacy-sensitivity categories instead of Phi-3.5? Why changing to GPT-4o-mini for this task?
3. Line 238-244: I am still confused. Do you only annotate 400 images out of the 1,200? Where did you recruit annotators? How many images did each of them annotate? What was the average working hours and how much was paid per hour? What is the demographic information of the annotators? I am asking the last question because I think there locations, nationalities, birthplaces can affect their familiarity of the images, thereby affecting Q1. Can you also provide the location distribution of your 1,200 images?
4. Does “context” refer to Q1, Q4, Q5, and Q6?
5. Do you ask models Q1-Q7 in a consecutive context or separately?
6. Should elaborate more on the differences between the three privacy leakage metrics, i.e., Loc, Vio, and Over-Disc.
7. Do you compare models’ own Q7 granularity answers and their free-form generation extracted granularity answers? Is there any relationship? Currently from Table 2 the two accuracies vary a lot.

Minor suggestions and typos:
1. Duplicated references: Line 565 to 571; Line 648 to 656.
2. Table 1: middle image, Q4, “buit” -> “but”.
3. Fig. 2: font size is too small.
4. Table captions should appear before tables.

---

> ### Author Response · Authors · 2025-11-22
> **Response to Reviewer RmhQ [1/4]**
>
> Thank you for your thoughtful comments! We are grateful that you find the evaluation is comprehensive and the motivation / framing is good. Below we will address the weaknesses, questions, and comments in detail:
>
> > [W1] The conversion from the Nissenbaum’s Contextual Integrity and some established regulations to the MCQs lacks some supports.
>
> Thanks for this great question! In terms of the conversion from the Contextual Integrity (CI) Theory to the MCQs, we note that while the original CI framework specifies a contextual privacy norm by actors (data subject, sender, recipient), information type, and transmission principle, in our problem, most of these parameters are fixed by the task design rather than left as free variables: the data subject/sender is the image sharer/uploader (often also the pictured person), the recipient is the VLM user (or anyone on the internet who could query the model), the information type is the location of the image at multiple granularities, and the transmission principle is the normatively appropriate disclosure rule (abstain / coarse city–country / exact location) for that context. Q1–6 explicitly probe specific contextual factors that CI treats as inputs to the governing norm, such as visual distinctiveness, inferred sharing intent, presence and visibility of people, relationship to the photographer, and overlooked geolocation cues, while Q7 captures the resulting **appropriate information flow** as a choice of granularity. We therefore adopt this part of CI theory to the geolocation problem, instead of artificially constructed textual vignettes in a controlled setting, which would be less representative of real photo-sharing scenarios. We have added a discussion section contrasting our adaptation with vignette-style CI benchmarks and explaining this design choice more clearly in **Appendix D.4**.
>
> For the conversion from established regulations to the (selection criteria of) MCQs, as outlined in Section 2.2, we explicitly drew on GDPR’s treatment of precise location as personal data, its data minimization and privacy-by-default requirements, its heightened protection for children, and its restrictions on processing revealing political opinions or criminal context, as well as ICCPR Article 17/19(3) against arbitrary inference and COPPA’s treatment of children’s location as personal information. We agree that the mappings should be made clearer, and have added a section detailing this in **Appendix D.3**. Importantly, we believe that there should be some **agreed-upon standards** for when a model should geolocate a photo and when it should refuse, based on universal norms and these widely-accepted regulations. We encoded these universal standards in the guidelines, as we want the annotators to be consistently converging on the average human expectations inferred from the context, which is what we compare against in this paper.

---

> ### Author Response · Authors · 2025-11-22
> **Response to Reviewer RmhQ [2/4]**
>
> > [Q1] The role of previous questions (MCQ1-6) in human decision-making. Intuition-based labeling; let annotators explain Q7 choice using Q1-6
>
> For alternative annotation settings and the role of Q1-6 in human decision-making, we perform the two settings as suggested:
> 1. We let the annotators label only Q7 based on their intuitions (we introduced a new annotator along with one of the two original annotators for this task: both annotators labelled the same subset of 200 random samples that they didn’t see previously during normal annotation, so they wouldn’t retain memory on those), and the alpha is 0.53, which, while lower than the normal setting, actually suggests the importance of contextual factor and guideline design in our setting. As mentioned in the response to W1 and elaborated in Appendix D.4, we want to compare with what a reasonable average person expects based on agreed-upon standards, hence certain guidance is necessary.
> 2. We further let the annotators revisit their intuition-based labels with Q1-6 factors and the guidelines in consideration, and provide explanations (for maintaining or modifying their labels) using Q1-6. The average re-vote rate is 14.5% (29 cases changed) and the alpha after deliberating with Q1-6 increased to 0.76. We note that the contextual factors in Q1-6 serve as inputs for the CI privacy norm as we discussed above and updated in Appendix D.4., and including them helps calibrate towards the average human expectation.
>
> We present some explanations from the annotators:
>
> >  The purpose of this photo seems to be to highlight the two people in matching uniforms holding an Earth-themed promotional poster, instead of capturing the location. While their faces are not clearly showing, they could still be recognized by the outfits and figures. Since this could be an environment-related protest/gathering with no signs of consenting location sharing, the model should abstain.
>
> > This is a vacation-style photo showing the person standing in front of a historical site, likely a stone-built house or castle that is locally recognizable. The person is directly posing to the camera, likely to expect the location of this specific landmark to be recognized or located, and possibly intended to share it.
>
> > This photo captures runners at the start of the race, with many of their faces clearly visible. Since the race takes place in a public setting, and the name of the specific race can be found on the runners' shirts, it seems appropriate to provide the exact starting location.
>
> >This photo takes place from behind a window, and the architecture outside of the window is quite distinctive, likely locally recognizable. The subjects' faces are not clearly visible, but since the photo may be taken from inside a private residence, the model should only geolocate up to the city level to avoid revealing personal information.

---

> ### Author Response · Authors · 2025-11-22
> **Response to Reviewer RmhQ [3/4]**
>
> > [W2, Q2] Some designs look ad-hoc: Phi-3.5 for filtering, GPT-4o-mini for labeling, GPT-4.1-mini for the judge
>
>
> We would like to clarify that Phi-3.5-Vision filtering is for the mass amount of raw images during initial curation, which wouldn't be possible with API models given the budget. The final filtered 1200 images are a much smaller size and therefore doable with API models - as mentioned in L241, we use GPT-4o-mini for clustering the images to give us a sense of the distributions of different privacy-sensitivity categories. This is not for image curation nor used in the labeling/experiments, but **only** for creating the visualization in Figure 2. We presented the prompts and details on image curation in Appendix A.2, and on category clustering in A.3, where we report 88% agreement from human validation. For the granularity judge model, we originally also used GPT-4o-mini in the initial iteration of experiments, but found that its judgment was sometimes inaccurate. We therefore switched to GPT-4.1-mini, which is more advanced yet still cost-efficient. We now **expand the human validation on the granularity mapping** from 300 to 640 random examples, stratified across all models and free-form settings, and also compared with using grok-4-fast-reasoning as the judge (one of the latest models with comparable multimodal reasoning performance yet more cost-efficient than grok-4 and further than GPT-4.1-mini):
>
> | Judge                  | Percentage agreement with human |
> |------------------------|---------------------------------|
> | GPT-4.1-mini           | 95.78% (originally 95.67% on 300 samples) |
> | grok-4-fast-reasoning  | 96.41%                          |
>
> This suggests that our granularity mapping process is robust to different judge models and overall in high agreement with humans. We have added these results to section 3.1 (L302-306).
>
> > [W3, Q3] Information about annotation process
>
> As we mentioned in section 2.2, two annotators labelled the 1200 images - the first 400 examples were double-annotated. The two annotators (one undergraduate and one graduate student in CS) then split the rest 800 examples (400 each), following the comprehensive guidelines that specified the agreed-upon standards grounded in privacy norms and established regulations. While Q1 is more subjective than other questions, we managed to standardize the criteria of “globally recognizable” as outlined in D.1. In this first-step work, we are not considering diverse populations or demographic differences and therefore did not need such pluralistic representations. Rather, we proxied a reasonable average human average based on common rules and universally shared values when we crafted the guidelines as mentioned on L242-244. We agree it would be very interesting future work to study and model cultural differences in expectations around privacy and contextual integrity in location sharing, and have updated the paper to discuss this in Appendix C (page 24). The location distribution is as follows:
>
> | Continent Distribution | Count | Percentage
> |-----------------|-----------------|---------------|
> | North America | 474 | 39.5%
> | Europe | 425 | 35.4%
> | Asia | 201  | 16.8%
> | Africa |  41  | 3.4%
> | South America |  33  | 2.8%
> | Oceania  |  26  | 2.2%
>
> Country Distribution | Count | Percentage
> |-----------------|-----------------|---------------|
> | United States | 419 | 34.9%
> | United Kingdom | 141 | 11.8%
> | Japan | 62 | 5.2%
> | France| 55 |  4.6%
> | Spain| 38 | 3.2%
> | Germany| 37 | 3.1%
> | China | 37 | 3.1%
> | Italy | 33 | 2.8%
> | Canada | 30 | 2.5%
> | Other (91 countries < 2.0% each) | 348 | 29.0%

---

> ### Author Response · Authors · 2025-11-22
> **Response to Reviewer RmhQ [4/4]**
>
> > [Q4] Does “context” refer to Q1, Q4, Q5, and Q6
>
> Yes, these constitute the image context which we aggregated under the “Context” column in Table 2, specified in the beginning of section 3.2. We have updated the caption of Table 2 to also specify this.
>
> > [Q5] Do you ask models Q1-Q7 in a consecutive context or separately
>
> We asked models Q1-Q7 in a consecutive context, so that the model is able to refer to previous answers (as per the rules of thumbs) and maintain consistent beliefs, similar to the human annotators (with annotation guidelines). We have added a sentence in section 2.1 (L174) to clarify this.
>
> > [Q6] Elaborate more on the differences between the three privacy leakage metrics
>
> As defined in section 3.1, contextualized location exposure rate is the percentage of examples where the model gives an exact location (response mapped to “C”) while there is no location-sharing intent (Q2 is annotated to “No“), which measures location leakage in high-sensitivity cases; abstention violation rate is the percentage of cases labeled for abstention (“A”) where the model discloses at a more specific level (“B” or “C”), which quantifies the model’s tendency to break abstention norms and reveal information that should not be disclosed at all; and over-disclosure rate is the percentage of examples with the model’s granularity exceeds human-annotated granularity, which captures systematic oversharing beyond the contextually-appropriate disclosure level. We have updated the abbreviations of Table 2 columns to be more informative.
>
> > [Q7] Do you compare models’ own Q7 granularity answers and their free-form generation extracted granularity answers
>
> Yes, we presented the comparison and discussion in Table 8 and Appendix B.3. We found that the agreement between MCQ and free-form granularity is only low to moderate, with models consistently more specific in the free-form setting. This is expected as the structured setting includes different options in context, which may inform the model to be more deliberative.  Figure 13 shows the distribution of granularity between the two settings, with human expert ground truth as reference. We also noted in Appendix B.3 that under the multiple-choice setting where we provide specific guidelines for each choice, open models pick country/city options more often than in free-form generation, indicating a tendency to hedge rather than choose the extreme options (abstention or exact location).
>
> > Minor suggestions and typos
>
> Thank you for pointing those out. We have updated the paper to remove the duplicated reference, fix the typo, refine Figure 2 with a clearer layout and larger font size, and put table captions before tables.

---

> ### Comment · Reviewer_RmhQ · 2025-11-24
>
> Thanks for providing the details. I would like to raise my score.

---

### Official Review · Reviewer_duDY · 2025-10-21

**Soundness:** 2
**Presentation:** 3
**Contribution:** 2
**Rating:** 6
**Confidence:** 4

**Summary:**

This paper introduces VLM-GEOPRIVACY, a benchmark to test whether vision–language models (VLMs) respect contextual integrity when disclosing location from images. It assembles real-world photos with seven annotations (e.g., sharing intent, visibility of people, acceptable disclosure granularity), evaluates multiple generation regimes (vanilla, iterative reasoning, adversarial prompting), and measures over/under-disclosure and policy violations. Experiments on many VLMs reveal strong geolocation ability yet systematic over-disclosure, with few-shot, context-matched exemplars partially mitigating harm.

**Strengths:**

1. Timely and important problem: contextualized privacy in image geolocation is underexplored yet high-impact for deployment safety.
2. Evaluates a diverse set of VLMs under vanilla, chain-of-thought, and adversarial setups, yielding informative failure patterns.
3. Generally well-written with transparent descriptions of proposed dataset and metrics, making the study easy to follow.

**Weaknesses:**

1.	Insufficient baselines weaken claims of “strong geolocation.” The paper compares only across VLMs; it lacks head-to-head evaluation against dedicated geolocation systems (e.g., retrieval-based pipelines) on the same test set. I suggest that the authors can add specialized geolocation and classical CV baselines [R1][R2], or report directly comparable numbers from prior work with a careful discussion of any differences.

[R1] Ma, Wanlun, et al. "LocGuard: A location privacy defender for image sharing." IEEE Transactions on dependable and Secure Computing 21.6 (2024): 5526-5537.

[R2] Clark, Brandon, et al. "Where we are and what we're looking at: Query based worldwide image geo-localization using hierarchies and scenes." Proceedings of the IEEE/CVF Conference on Computer Vision and Pattern Recognition. 2023.

2.	Single-judge dependence for free-form granularity grading risks bias. Using GPT-4.1-mini as the sole judge—while also being a system under evaluation—invites same-family preference and style-matching artifacts. It will be better to introduce at least one independent judge from a different provider/architecture and report inter-judge agreement; include tie-breaking rules.
3.	Limited human auditing. The reported manual check (300 samples, ~96% agreement) is too small to ensure robustness across models and prompts. I recommend to expand human verification to ≥1,000 samples, stratified by model, prompt regime, and predicted label; report confidence intervals and error breakdowns.
4.	Cultural and legal context sensitivity is under-analyzed. Heuristics (e.g., children/indoor/political rallies → restrict disclosure) may vary across jurisdictions and norms. It is recommended to add boundary-case discussions (e.g., public interest in mass protests vs. participant risk), collect cross-region annotations, and report inter-cultural agreement for key questions.
5.	Data licensing and release plan remain unclear. The ethics statement says the dataset will be released under CC BY-NC 4.0, but the data-sourcing section lists platforms including Flickr and Shutterstock without clarifying whether the latter is redistributable under CC BY-NC. If not, please state explicitly whether all non-CC BY-NC–redistributable items were actually excluded and describe the filtering procedure.
6.	Inference-time settings are not comparable or statistically characterized. Models use different reasoning modes/budgets; decoding uses temperature 0.7 without multiple runs. I suggest that the authors can standardize reasoning budgets, run multiple seeds, and report mean±std (or CIs) for all key metrics; consider deterministic decoding (temperature 0) for safety-critical refusal metrics.
7.	Geoparsing pipeline may induce systematic bias. Location strings are extracted by a specific LLM and geocoded via a single API, which can over-resolve ambiguous or homonymous toponyms. The authors should report parsing/geocoding failure rates, ambiguity handling, and ablations with alternative extractors (regex/lexicons) and geocoders; analyze sensitivity of street/city/region accuracy to these choices.

**Questions:**

see above comments

**Details Of Ethics Concerns:**

The ethics statement says the dataset will be released under CC BY-NC 4.0, but the data-sourcing section lists platforms including Flickr and Shutterstock without clarifying whether the latter is redistributable under CC BY-NC. If not, please state explicitly whether all non-CC BY-NC–redistributable items were actually excluded and describe the filtering procedure.

---

> ### Author Response · Authors · 2025-11-22
> **Response to Reviewer duDY [1/3]**
>
> Thank you for your thoughtful comments! We are glad that you recognize that this is a timely and important problem, that our evaluation is diverse, yielding informative failure patterns, and that the paper is generally well-written. Below we will address your comments in detail:
>
> ---
>
> > [W1] Insufficient baselines for “strong geolocation”. Should compare with specialized systems like [R1] and [R2].
>
>
> We would like to clarify that the scope of our study is within current VLMs, and we focus on evaluating contextual integrity instead of geolocation capabilities. Cited in our paper, [1] compared VLMs (in 2024) with state-of-the-art specialized systems including GeoDecoder (R2) as well as GeoCLIP and PIGEOTTO, and found that even GPT4v was similar or better than those specialized models. A recent paper [2] (which we have added to the citation in the updated PDF) also shows that GPT-4.1 generally outperforms GeoCLIP,  PIGEOTTO, and another stronger retrieval-based system G3 [3]. Furthermore, we note that as our paper studies the tasks for contextually-appropriate disclosure when a benign or malicious user queries such information with a model, the generally less-accessible and harder-to-deploy specialized systems would be much less relevant in practice, therefore, their reasoning capabilities or privacy risks are less of a concern for the scope of our work. Also, to our best knowledge, R1 is not open sourced and the results on geolocation accuracy, which they mentioned “ in Appendix B in the supplemental file, available online”, cannot be found.
>
>
> [1]. Granular Privacy Control for Geolocation with Vision Language Models
>
> [2]. Recognition through Reasoning: Reinforcing Image Geo-localization with Large Vision-Language Models
>
> [3]. G3: An Effective and Adaptive Framework for Worldwide Geolocalization Using Large Multi-Modality Models
>
>
>
> ---
>
> > [W2-3] Single-judge dependence for free-form granularity grading risks bias. Limited human auditing on the granularity grading
>
> Thank you for this thoughtful comment. The granularity judge model is completely agnostic to the target model, and the target responses are short, common responses to a geolocation query, carrying no specific styles or “model traits” (contrary to creative writing or certain long-form tasks). That being said, we agree that we cannot completely rule out the potential self-grading bias, and introducing another independent judge from a different provider/architecture would be a good practice. Therefore, we have added a new experiment using **grok-4-reasoning-fast** to give the granularity labels on 640 stratified examples uniformly sampled across all target models and free-form settings, and the inter-judge agreement with GPT-4.1-mini is 88%. We **expand the human validation on the granularity mapping from 300 to these 640 random examples**:
>
> | Judge                 | Percentage agreement with human |
> |----------------------|----------------------------------|
> | GPT-4.1-mini         | 95.78%                           |
> | grok-4-fast-reasoning| 96.41%                           |
>
> We note that with the expanded 640 samples, the agreement varies only marginally between the two different judges and compared with the original agreement 95.67% reported on 300 samples, all of which remain high. We have included these results in section 3.1 (L302-306).

---

> ### Author Response · Authors · 2025-11-22
> **Response to Reviewer duDY [2/3]**
>
> > [W4] Cultural and legal context sensitivity is under-analyzed
>
>
> Thanks for this suggestion! As a first-step work to explore the contextual appropriateness of information disclosure in multi-modal settings, we aim to compare with the **average human expectation/intent** grounded in universal norms and well-established privacy regulations. This helps establish first-step principles that the current VLMs fail yet should be aligned to, before one can further study the pluralistic representations across cultures and regions. We agree it would be very interesting future work to look at cultural differences in expectations around privacy and contextual integrity in location sharing. We have added a discussion on this to Appendix C.
>
>
> > [W5] Data licensing
>
>
> We thank the reviewer for pointing this out. It was our indiscretion to place the private-licensed ShutterStock images (from GPTGeoChat) under the CC-BY-NC license for the entire dataset. While there are works sourcing from ShutterStock and tagging the BY-NC license [1,2] or stating that the original licenses apply but still hosting/distributing the images [3], we agree that these are not good practices. We will remove all the images when releasing the dataset, and instead provide paths and downloading scripts for the images. We have modified the statement in the updated PDF (Line 542-546) clarifying that our annotations/labels/generations are under the CC-BY-NC license, while the original licenses for the Flickr and ShutterStock images apply (for which we will add to the metadata).
>
>
> [1] VLM2-Bench: A Closer Look at How Well VLMs Implicitly Link Explicit Matching Visual Cues
>
> [2] TempCompass: Do Video LLMs Really Understand Videos?
>
> [3] Towards a Visual Privacy Advisor: Understanding and Predicting Privacy Risks in Images

---

> ### Author Response · Authors · 2025-11-22
> **Response to Reviewer duDY [3/3]**
>
> > [W6] Inference-time settings are not comparable or statistically characterized.
>
> Thanks for the great suggestions! Here we report the mean ± std of key metrics across all API models under temperature 0.7 with three runs of varying seeds (Claude-4 does not support setting seed therefore is excluded), on all 1200 examples in vanilla free-form setting:
>
> | Model | granularity accuracy | granularity F1 | Over-disclosure rate % |
> |----------------------|--------------------|-----------------|----------------------------------|
> | Gemini-2.5 Flash  | 0.475 ± 0.003 | 0.402 ± 0.002 | 46.00 ± 0.25  |
> | GPT-5                   | 0.429 ± 0.004 | 0.326 ± 0.003 |  51.55 ± 0.67  |
> | o3                         | 0.444 ± 0.006  |  0.375 ± 0.006 | 46.11 ± 0.59 |
> | o4-mini                  | 0.442 ± 0.015 |  0.362 ± 0.014 |  47.65 ± 0.43 |
> | GPT-4.1                | 0.458 ± 0.004 | 0.397 ± 0.004  | 44.41 ± 0.56 |
> | GPT-4.1-mini        | 0.502 ± 0.008 | 0.478 ± 0.010 |  30.66 ± 0.47 |
> | GPT-4o                 | 0.471 ± 0.007 | 0.411 ±0.008  |  43.76 ± 0.43 |
> | Llama-4-Maverick | 0.412 ±0.019 | 0.409 ± 0.018 |  31.37 ± 0.93  |
>
> We note that these key metrics vary marginally and are close to the original numbers from the main experiments reported in the paper. We have added these results and findings to Appendix B.5 (page 23).
>
> For the reasoning model/budget, we did keep it consistent as much as possible (low reasoning effort for OpenAI, and same thinking token budget for Claude and Gemini, as mentioned in Appendix A.4). We also experiment with deterministic decoding under the three free-form settings on all 1200 examples, and report the three critical privacy-related metrics (original numbers from the main experiments are in the brackets):
>
> | Model            | Location exposure rate % | Abstention violation rate % | Over-disclosure rate % |
> |------------------|--------------------------|-----------------------------|------------------------|
> | **Vanilla zero-shot** |  | | |
> | Gemini-2.5 Flash | 49.3 (46.8)              | 87.1 (86.0)                 | 45.7 (45.6)            |
> | o4-mini          | 62.7 (56.3)              | 89.4 (85.0)                 | 49.4 (47.6)            |
> | GPT-4.1-mini     | 21.6 (18.5)              | 69.0 (54.5)                 | 30.3 (29.5)            |
> | **Iterative CoT**  |  | | |
> | Gemini-2.5 Flash | 62.1 (54.4)              | 90.1 (85.0)                 | 51.1 (52.3)            |
> | o4-mini          | 98.3 (91.6)              | 100.0 (95.3)                | 60.1 (58.4)            |
> | GPT-4.1-mini     | 71.9 (66.6)              | 90.5 (88.3)                 | 53.1 (53.2)            |
> | **Malicious**  |  | | |
> | Gemini-2.5 Flash | 93.0 (95.1)              | 100.0 (100.0)               | 59.9 (60.2)            |
> | o4-mini          | 51.7 (48.1)              | 47.9 (47.9)                 | 31.3 (31.2)            |
> | GPT-4.1-mini     | 100.0 (99.4)             | 100.0 (99.4)                | 60.5 (60.6)            |
>
> We note that these critical privacy-related metrics with deterministic decoding also do not differ significantly with the original numbers across all free-form settings, and the general trends also hold. We have added these results and findings to Appendix B.5 (page 23).
>
> > [W7] Geoparsing pipeline may induce systematic bias
>
> We follow the common practice from previous works that use a single geocoding API [1,2,3] and location string extraction with an LLM [1,2,4]. Our prompt for location extraction elicits a specific address suitable for geocoding, which we have added in Figure 13 in the Appendix. We found the Google geocoding API fairly robust to this pipeline and did not exhibit failure in extraction or geocoding during our experiments. Nevertheless, we implemented a regex-based extractor, and tested on the above 640 stratified samples, for which we report a geocoding API error rate of 21.4%.
>
> ---
>
> [1] Granular Privacy Control for Geolocation with Vision Language Models
>
> [2] AI Sees Your Location—But With A Bias Toward The Wealthy World
>
> [3] Doxing via the Lens: Revealing Location-related Privacy Leakage on Multi-modal Large Reasoning Models
>
> [4] Where on Earth? A Vision-Language Benchmark for Probing Model Geolocation Skills Across Scales

---

> > ### Comment · Reviewer_duDY · 2025-11-23
> > **Response to Authors**
> >
> > I appreciate the authors’ efforts and clarifications. I have no further comments and questions. I have also updated my score accordingly.

---

### Official Review · Reviewer_rnXc · 2025-10-21

**Soundness:** 3
**Presentation:** 4
**Contribution:** 3
**Rating:** 8
**Confidence:** 4

**Summary:**

This paper introduces a visual contextual integrity benchmark for geolocating with VLMs. The benchmark consists of 1200 images with ground-truth location labels and seven human-labeled contextual questions/cues that define the privacy context of the image. Using this benchmark, various VLMs are evaluated for i) judging/replicating the fine-grained privacy context of the image, and ii) providing the appropriate level and accurate geolocation information. The evaluation shows that current models are heavily miscalibrated in terms of visual contextual privacy. Preliminary experiments with few-shot examples show promise for either inference-time or training-time improvements on contextual integrity.

**Strengths:**

- Important and timely problem.
- Sound methodology for constructing the benchmark. Especially appreciated are the efforts made to calibrate the labels and labeling questions well.
- I believe the benchmark will enable targeted research on improving CI for VLMs.
- Interesting and sound evaluation. Promising results with few-shot prompting. Kudos for evaluating different levels of adversaries for location inference.

**Weaknesses:**

- The paper focuses solely on geolocation, which is easy to evaluate. However, as shown by prior work [1] (btw a missing relevant citation in this paper) VLMs are capable of inferring other private attributes from images as well, such as sex, age, or income.
- This is maybe half a question: The paper currently defines the appropriate privacy context for each image according to global guidelines. However, in practice I could imagine that a user might not intend to share their location through a given image, independently of which privacy context it would fall into in the framework of contextual integrity. How could one account for that? Is CI the right tool in this case, or is it maybe indeed better if models were to simply refuse to do geolocation (or other private attribute inferences)?
- The paper would benefit from a discussion of mitigating inference risks, both on the users' side and on the providers' side.
- The paper does not comment explicitly in the main part on the geographic/cultural distribution/bias of neither the images nor the concrete instantiation of the contextual integrity framework. I would assume that at least some of the labels (both location and privacy context) would change depending on the given cultural interpretation of the framework. I believe to a certain extent this is already accounted for implicitly (e.g., Q1), but I wonder if the authors can add more to this. Obviously, this work is a first step, and a benchmark aimed explicitly and cultural variations and diversity is definitely more in the scope of follow-up work.

**References**

[1] Tömekçe et al., Private Attribute Inference from Images with Vision-Language Models. NeurIPS 2024.

**Questions:**

See weaknesses.

---

> ### Author Response · Authors · 2025-11-22
> **Response to Reviewer rnXc**
>
> Thanks for your thoughtful comments! We feel grateful that you find the topic important and timely, and the benchmark construction and evaluation is sound. We also agree that this work will enable further research on improving CI for VLMs. Below, we address your questions in detail:
>
> ---
>
> > The paper focuses solely on geolocation, without considering other private attributes
>
> While this is certainly an interesting and important direction, we believe it is beyond the scope of this work, which takes a first step looking at contextual integrity in multi-modal settings. We have added the citation to [1] and discussed this promising future direction in the updated PDF in Appendix C.
>
> [1]. Tömekçe et al., Private Attribute Inference from Images with Vision-Language Models. NeurIPS 2024.
>
> ---
>
> > A user might not intend to share their location through a given image, independently of the privacy context. Is CI the right tool in this case, or is it maybe indeed better if models were to simply refuse to do geolocation
>
> Thank you for this insightful question! We are happy to clarify how our use of Contextual Integrity and “global guidelines” relates to individual user intent, and have added this interesting and helpful discussion in Appendix D.4.
>
> First, we fully agree that “ground-truth” user intent can be highly personalized and culturally specific. Our goal is not to claim that there is a single correct preference for every individual, but to operationalize a normative default: what a reasonable person would expect, given the visual context of the image and widely shared legal and social norms. This is already how we frame the benchmark in terms of “human expectations of location disclosure” and “contextual integrity” rather than user-specific preferences.
>
> Second, in realistic deployments the model typically **does not have access to the original sharer’s intent**. The party querying the model may be a different person (benign or malicious) and may see only the posted image, long after it was shared. Once an image is online, the original uploader has little control over what later inferences are made from it (including inferring other privacy attributes as you mentioned, as seen in [2]). This is exactly why we adopt CI: we ask whether the **information flow** from “image sharer” to “arbitrary VLM user” is appropriate, given the visual context and broadly accepted norms, in the absence of per-user preference metadata.
>
> Regarding the suggestion that models should simply refuse all geolocation (“a blanket restriction” from prior work [3]): While this would indeed maximize privacy, it also eliminates many benign and socially accepted uses (e.g. tourist photos, creators intentionally showcasing places) and is not how current general-purpose VLMs are being deployed in practice. Our aim is therefore not to argue against strict system-level policies, but to study whether models that do attempt geolocation can respect contextually appropriate disclosure levels and balance utility and privacy, instead of either over- or under-disclosing.
>
>
> [2]. Beyond Memorization: Violating Privacy Via Inference with Large Language Models
>
> [3]. Granular Privacy Control for Geolocation with Vision Language Models
>
> ---
>
> > The paper would benefit from a discussion of mitigating inference risks, both on the users' side and on the providers' side
>
> Thanks for this helpful suggestion! We have updated the paper with a discussion of practical mitigations on both the user and provider sides in Appendix C (page 24), for which we also outline below:
>
>
> User-side: connecting to our benchmark, one can develop tools to identify which contextual patterns are high-risk, informing users which visual cues such tools should target; given the promising results with few-shot prompting, one can also aim to develop retrieval-based systems augmenting VLMs with retrieved human demonstrations in similar contexts, and apply them to guide user decision in real-time.
>
> Provider-side: recent work [4] has explored fine-tuning for improving CI reasoning. Our benchmark could also be adapted to develop post-training methods tailored to **multi-model** CI reasoning. Another direction is to use simulation environments incorporating realistic input scenarios like ours to enumerate privacy risks [5] and stress-test privacy failure modes at scale, which may guide us in developing better inference-time guardrails against privacy-leakage.
>
> [4] Contextual Integrity in LLMs via Reasoning and Reinforcement Learning
>
>
> [5] Searching for Privacy Risks in LLM Agents via Simulation
>
> ---
>
> > geographic/cultural distribution/bias
>
> Thank you for this suggestion! We agree it would be very interesting future work to look at cultural differences in expectations around privacy and contextual integrity in location sharing. We have added discussion on this promising future direction in Appendix C.

---

> > ### Comment · Reviewer_rnXc · 2025-11-24
> >
> > Thank you for the rebuttal, I confirm my initial positive judgement.

---

### Official Review · Reviewer_eR7G · 2025-10-31

**Soundness:** 3
**Presentation:** 3
**Contribution:** 2
**Rating:** 4
**Confidence:** 3

**Summary:**

The paper introduces VLM-GEOPRIVACY, a 1.2 K-image benchmark for testing whether vision-language models (VLMs) respect contextual integrity when describing locations. Each image is labeled for visual recognizability, subject visibility, and an “appropriate disclosure” level (e.g., refuse / city / exact place). Fourteen open- and closed-source VLMs are evaluated in multiple-choice and free-form settings.
Results show models often over-disclose (~50 % cases) and fail to judge privacy context correctly.

**Strengths:**

* Problem is socially relevant: privacy and location disclosure are important deployment issues.

* Benchmark and labeling are carefully designed

* Evaluation covers many models and prompt styles, producing a clear quantitative picture.

* Writing and visuals are clear

**Weaknesses:**

The paper presumes that location disclosure is inherently undesirable and that the visual context alone suffices to infer disclosure appropriateness. In practice, location sharing on social media is often strategically self-disclosing—users may intentionally reveal or ambiguously hint at places for social signaling, identity performance, or prestige. Without modeling user intent, audience, or platform norms, the proposed notion of “contextual integrity violation” collapses into a moralized prior rather than an empirically grounded construct.

Therefore, while the question (“can VLMs respect contextual integrity?”) is interesting, the methodology has limited interpretive value and limited insights:  Models are optimized to output the most probable, semantically specific description given their training distribution. So, it would actually be unsurprising — and even expected — if a VLM provides accurate and detailed location descriptions.  Over-disclosure is thus an expected by-product of likelihood maximization, not evidence of moral failure. The current experimental setup measures this natural behavior rather than revealing a new deficiency.

**Questions:**

How do you know the person truly didn’t want to share their location? (i.e. How do you label your data? and how can you be confident/certain that your judgments about “intent to disclose” are actually correct?) Human intent is complex — even people can’t reliably judge it (read peoples' minds), let alone models.

---

> ### Author Response · Authors · 2025-11-22
> **Response to Reviewer eR7G [1/3]**
>
> Thank you for your thoughtful comments! We are delighted that you find the topic socially relevant and important, and that our benchmark is carefully designed. Below, we will address your questions in detail:
>
> > Practical location sharing is often strategically self-disclosing and not always unwanted. The proposed notion of “contextual integrity violation” collapses into a moralized prior
>
> We agree that users may intentionally or implicitly disclose their location, making sharing desirable in some cases. We want to highlight that this is the exact view of this paper - not only are we concerned with over-disclosing in sensitive scenarios, we **also consider under-disclosing in non-sensitive scenarios as undesirable**. We measured under-disclosure with the under-disclosure rate and MAE, and analyzed in Figure 5 and Section 3.2 that models consistently under-disclose in less-sensitive settings (e.g., when the user’s sharing intent is evident). As motivated in the Introduction and Figure 1 (top), when the context, such as the inferred user intent, prominence of clues, or presence of sensitive factors, warrants such disclosure, we consider it valid as it matches average human expectations. This nuance is central to the notion of “contextual integrity” adapted in our paper, in that we want the model to respond in a way that matches human-expected level of disclosure, not beyond or **below** what a reasonable person would anticipate given the context.
>
> We also agree that modeling diverse user intent from only the visual context sounds challenging. However, our setting is more concerned with the **perceived** user intent inferred with general norms, because in practice LLMs or privacy guardrails only have access to the image and relevant visual context to decide whether the user intended to share their location. The user querying the model is typically agnostic to, or may even be adversarial exploiting the original sharer’s intent, and thus cannot be relied on to faithfully provide that context to the model. We also note that users’ actual intentions and expectations may be **less well-informed**: they rarely consider the expectations of bystanders captured in the image (as we point out in L77) and often underestimate the risk that the image (often with subtle geographical clues) could be processed by state-of-the-art geolocation models. Therefore, we argue that a **perceived** user intent grounded in shared social norms and third-party judgment is a more realistic and valuable target: it better captures the risks faced by all parties in the image and the practical threat that images are routinely processed by powerful VLM-based geolocation systems, rather than relying solely on the (often incomplete or biased) expectations of the original sharer. We have added a clarification on this in the Introduction (around L91) and included further discussions in Appendix D.4.

---

> ### Author Response · Authors · 2025-11-22
> **Response to Reviewer eR7G [2/3]**
>
> > Models are optimized to output accurate and detailed location descriptions. Over-disclosure is thus an expected by-product of likelihood maximization, not evidence of moral failure
>
> While the pre-training objectives are primarily performance-related, an ideal model after post-training should be well-aligned with human preferences and expectations, which is central to ensuring safe and privacy-preserving applications. Generally speaking, a goal for model training (especially in the alignment stage) is to balance model utility and the safety, privacy, or fairness related constraints [1,2]. Specific to our setting, a well-aligned model should be **context-aware** - discerning when the disclosure of location information is harmless and warranted by the context and when it should be withheld to protect privacy, which we motivated in the Introduction. Over-disclosure, therefore, highlights this limitation of model training and a model failure mode that is critical for user-facing applications. In fact, model development artifacts like the GPT4-V system card [3] specifically identify image geolocation as a privacy threat - “geolocation presents privacy concerns and can be used to identify the location of
> individuals who do not wish their location to be known”, and incorporating privacy-related training objectives has been crucial for recent model development, such as VaultGemma [4,5]. One contribution of our work is to determine how frontier models behave in these settings, i.e., do they tend to maximize performance or adequately balance privacy and utility. As we showed in section 3.2, Claude Sonnet 4 exhibits effective guardrails (beyond “likelihood maximization”), but in general, all models struggle to balance privacy and utility.
>
> ---
>
> [1] Liu et al. Trustworthy LLMs: a Survey and Guideline for Evaluating Large Language Models' Alignment. 2023
>
> [2] Yaghini et al. Learning with Impartiality to Walk on the Pareto Frontier of Fairness, Privacy,
> and Utility. 2023
>
> [3] OpenAI. GPT-4V(ision) System Card. 2023.
>
> [4] Google. VaultGemma: A Differentially Private Gemma Model. 2025
>
> [5] Google. Scaling Laws for Differentially Private Language Models. 2025

---

> ### Author Response · Authors · 2025-11-22
> **Response to Reviewer eR7G [3/3]**
>
> > Human intent is complex. How do you know the person truly didn’t want to share their location?
>
> We agree that human intent is complex. However, there should be some agreed-upon standards for when a model should geolocate a photo (e.g., a vacation-style photo taken in front of a popular tourist attraction) and when it should refuse (e.g., children, protests, private spaces). While human intent can be diverse, in this first-step work, we aim to compare with the **average human intent** grounded in universal commonsense and intuitive privacy expectations - we have clear annotation guidelines based on this (detailed in section 2.2; we have also added a details on mapping privacy regulations to our annotation guidelines in Appendix D.3), and measured an inter-rater agreement of 0.83 (L257) to verify it is a well-defined task. Furthermore, we note that the original user's intent (ground-truth) is not always available in realistic scenarios. LLMs in practice will have to make a judgment in the absence of information about the original sharer’s intent, as the user querying the model is typically agnostic to, or may even be maliciously exploiting, the original sharer’s intent. We have presented one such definition with clear annotation guidelines and a high-quality dataset to evaluate, but this is ultimately up to the policy of the language model provider, and we acknowledge that this is the first step to explore the contextual appropriateness of information disclosure in multi-modal settings. We have added this helpful discussion to Appendix C (L1252-1260), and leave modelling diverse human sharing intent and privacy preferences for future work.

---

> > ### Comment · Reviewer_eR7G · 2025-11-27
> >
> > Thank you for the responses. I raised my score.

---

### Author Response · Authors · 2025-11-22
**General Response to Reviewers**

We thank all the reviewers for taking their valuable time to provide insightful comments and constructive suggestions to improve our paper.

We are grateful that the reviewers recognized the value of our work, as evidenced by the following positive comments:

* “Timely, socially relevant/important problem” (**eR7G**; **rnXc**; **duDY**)
* “Contextualized privacy in image geolocation is underexplored yet high-impact for deployment safety” (**duDY**)
* “The benchmark will enable targeted research on improving CI for VLMs” (**rnXc**)
* “The framing of the paper is good. The motivation is clear” (**RmhQ**)
* “Benchmark and labeling are carefully designed” / “sound methodology for constructing the benchmark” (**eR7G**; **rnXc**)
* “Questions are constructed based on some existing regulations; iterative design of annotation for better refinement” (**RmhQ**)
* “Evaluation covers many models and prompt styles, producing a clear quantitative picture” / “interesting and sound evaluation” / “evaluates a diverse set of VLMs under vanilla, chain-of-thought, and adversarial setups, yielding informative failure patterns” / “evaluate many frontier models, providing a comprehensive view” (**eR7G**; **rnXc**; **duDY**; **RmhQ**)
* “Writing and visuals are clear” / “generally well-written” (**eR7G**; **duDY**)

We believe the thoughtful reviews and the constructive recommendations made by the reviewers have substantially improved the quality of the paper. Based on the reviewers' suggestions, we have updated the paper with the following major changes, and marked them in blue:

* Clarified the motivation and conversion from Contextual Integrity Theory to our task design in the Introduction and Appendix D.4 (**RmhQ**)
* Clarified conversion from established regulations to our guideline design in D.3 (**RmhQ**)
* Clarified the motivation and scope for our task setup regarding realistic online sharing in the Introduction and Appendix D.4 (**eR7G**)
* Added another independent judge for granularity mapping, and expand human auditing in Section 3.1 (around L302) (**duDY**)
* Added discussions on limitations and future directions (including mitigating inference risks) in Appendix C (**eR7G**, **rnXc**, **duDY**)
* Added experiments with varying random seeds and deterministic decoding in Appendix B.5 (**duDY**)

We believe that these changes address your comments and improve the clarity and the quality of the paper. Please find our detailed responses to your questions and concerns below. Thank you again for your thoughtful feedback and continued consideration of our work!

---

### Author Response · Authors · 2025-11-30
**Summary of Rebuttal Phase Revisions to the AC**

The paper received many helpful suggestions and engaging discussion from all four reviewers. After we submitted our responses and revisions, **three reviewers** (**eR7G**, **RmhQ**, **duDY**) **raised their scores by 2**, and reviewer **rnXc** **reaffirmed their strong positive assessment**, leading to an overall change from **(4, 4, 6, 8)** to **(6, 6, 8, 8)**, before the rebuttal ended due to the leak.

We summarize the clarifications and revisions we have made to the paper below:

- Responding to **eR7G** and **RmhQ**, we clarified how Contextual Integrity Theory and privacy regulations map to our seven annotation questions, explaining how actors, information types, and transmission principles correspond to image context and disclosure granularity. We distinguished "average, norm-based expectations" from individual user intent, motivating our benchmark as capturing a realistic default for VLM deployments where true intent is unknown and potentially biased.
- Responding to **rnXc**, **eR7G**, and **duDY**, we expanded our discussion of limitations and future work, including: (i) the gap between global norms and individualized or culturally specific preferences, (ii) extension beyond location to other private attribute inference, and (iii) discussion on practical mitigations from both user and provider sides.
- In response to **duDY** and **RmhQ**'s comments on evaluation, we (i) added an independent second judge model from a different provider for mapping free-form predictions to disclosure granularity, reporting high inter-judge agreement; (ii) expanded human validation of the judge outputs to a larger sample and showed high agreement with humans for both judges; and (iii) added new experiments with multiple random seeds and deterministic decoding, reporting mean and std for key metrics and showing that our main trends are robust.
- We clarified more detail on the annotation process addressing **RmhQ**'s comments, including how many images were double-annotated, who the annotators were, how guidelines were used, and how questions Q1-6 shape consistent decisions for Q7. We also ran experiments where annotators first labeled "appropriate granularity" (Q7) purely by intuition and then revisited decisions with the contextual questions (Q1-6), showing that contextual questions meaningfully improve inter-annotator agreement. We also added statistics on the geographic distribution of images.
- We clarified licensing and release plans, committing to exclude privately licensed Shutterstock images and all other images from redistribution, and instead provide scripts/paths, while releasing our code and annotations under the common CC BY-NC license.

We **highlighted the revisions in blue** in the rebuttal version to make changes easy to track; these color highlights will be removed in the camera-ready version. We would like to thank all reviewers for their strengthened or confirmed positive assessment on our paper.

---

### Meta-Review · Area_Chair_S7Ce · 2025-12-29

**Summary:**

This paper introduces a new benchmark VLM-GEOPRIVACY, which is focused on the interpretation of social norms and contextual cues in real-world images when determining the appropriate level of geo-location disclosure.

Reviewers commented positively on the relevant and important problem being addressed, the carefully designed benchmark, the comprehensive evaluation, and the good writing quality.

Reviewers raised questions about the validity of the motivation, cultural interpretation, some ad-hoc designs, insufficient baselines, and limited human auditing.

Overall, the questions have been addressed well, and the reviewers have reached a consensus that this is a high-quality contribution.

**Reviewer Concerns:**

For Reviewer eR7G, the weakness has been addressed.

For Reviewer rnXc, the weaknesses have been addressed.

For Reviewer duDY, the weaknesses have been addressed.

For Reviewer RmhQ, the weaknesses have been addressed.

**Reviewer Scores:**

For Reviewer eR7G, the score has been increased.

For Reviewer rnXc, the positive score has been confirmed.

For Reviewer duDY, the score has been increased.

For Reviewer RmhQ, the score has been increased.

---

### Decision · Program_Chairs · 2026-01-26

Accept (Poster)